# Cross-sectional survey among professionals on communication and mental health care for asylum seeking and refugee minors in Germany

Anna Jael Esser [1], Jana Willems [1,2], Mia Klein[3], Markus Hufnagel[3], Thorsten Langer[1] & Benedikt D. Spielberger [3] ✉

## Abstract

**Background:** Forced displacement and migration are on the rise worldwide. Asylum seeking and refugee minors (ASRM) are particularly exposed to risk factors for mental health problems. Yet, there is a lack of comprehensive data on the prevalence of specific mental health problems as well as applied screening and follow-up care in Germany.
**Methods** Using the online platform REDCap, we conducted the cross-sectional SAVE-KID survey among health and social care professionals (HSCP) working with ASRM in Germany (n = 201; 44% medical doctors, 38% social workers) to assess the estimated mental health burden among ASRM, the conducted screening measures, and provided mental health care as well as the extent to which communication problems affect care for ASRM.
**Results** Here we show, that on average, 21% of ASRM are reported with one or more listed mental health problem. Only 37% receive follow-up. Less than 24% of participants conduct mental health screening by informal questions, interviews, trained staff or questionnaires. 84% of participants report frequent communication problems. Most used aids are online tools or relatives' translations.
**Conclusions** SAVE-KID describes an imbalance between the occurrence of and screening for mental health problems among ASRM. Comprehensive, systematic detection of mental health problems remains challenging due to communication problems, lack of specialized staff and diagnostic tools as well as follow-up care structures.

## Plain Language Summary

Children and adolescents who apply for asylum or refugee status in Germany are exposed to numerous mental health risks. We conducted an online survey called SAVE-KID, in which 201 health and social care professionals participated. Respondents reported that about one in five minors had a mental health problem, but only one-third of them received follow-up care. Less than a quarter of professionals used structured mental health screening methods. Most professionals reported serious communication difficulties and often had to rely on online tools or family members for translations. These findings show that mental health problems are widespread among underage asylum seekers and refugees, but effective screening and care are hampered by communication barriers, a lack of trained staff, and inadequate diagnosis and support structures.

Forced displacement and migration are on the rise worldwide. By mid-2024, 122.6 million people were forcibly displaced, including 72.1 million internally displaced[1]. Germany ranks among the top five receiving countries, hosting about 3.3 million displaced persons, including nearly 900.000 minors[2]. Main countries of origin currently are Syria, Afghanistan, and the Ukraine. Asylum seeking and refugee minors (ASRM) are over-represented with a share of 40% as compared to their global share of 30%. In

Germany, 16.9% of the general population but 31.5% of asylum applicants are minors[3]. In 2023, this equaled new arrivals of >320,000 minors (420,000 in 2015; 74,000 in 2005[4,5]). The group of ASRM is highly heterogeneous (age, legal status, origin, cultural family/social setting), but share increased risks for health and limited access to care[6].

Systematic reviews show high rates of mental health problems among ASRM: Post-traumatic stress disorder (PTSD; 19–52.7%), depression

[1]Department of Neuropediatrics and Muscle Disorders, Department of Pediatrics and Adolescent Medicine, University Medical Center Freiburg, Faculty of Medicine, University Freiburg, Freiburg, Germany. [2]Section of Health Care Research and Rehabilitation Research, Institute of Medical Biometry and Statistics, University Medical Center Freiburg, Faculty of Medicine, University Freiburg, Freiburg, Germany. [3]Division of Pediatric Rheumatology and Clinical Infectious Diseases, Department of Pediatrics and Adolescent Medicine, University Medical Center Freiburg, Faculty of Medicine, University Freiburg, Freiburg, Germany. ✉e-mail: benedikt.spielberger@uniklinik-freiburg.de

(10–32.8%), anxiety (9–32%), behavioural issues (20–35%)[7,8]. Yet, data are heterogenous, and recent epidemiological data in Germany are scarce. The German KiGGS/BELLA-Study found an overall prevalence of 17.2% in minors from German-speaking families screened based on the Strength and Difficulties Questionnaire (SDQ)[9] and additional questionnaires[10,11]. The post-pandemic COMO-Study reports mental health problems in 21% of German-speaking minors with an important impact of the socio-economic status[12].

For those with mental health care needs, multiple barriers in accessing care arise, as depicted by recent reviews[13,14]. Initially, an objective and perceived therapeutic need must be recognised, the former ideally through comprehensive screening. Current national screening guidelines mainly focus on infectious diseases[15]. While they do recommend assessing psychiatric disorders and stress factors, they do not specify adequate tools. Extensive questionnaires for different psychiatric disorders exist, yet these are often time consuming, require professional expertise, and are limited to sufficient resources and to a clear suspicion for a certain disorder. Furthermore, their validated implementation often exclude ASRM[16,17]. Two screening tools tend to be more widespread: The Refugee Health Screener (RHS-15; validated only >14 y, <20 languages)[18] and Strengths and Difficulties Questionnaire (SDQ; 2–17 y)[9]. The SDQ is freely available in > 80 languages as parent, teacher and self-rates versions. Years of broad application of the SDQ in practice as well as in science offer points of reference and comparison. Internal consistency is reported with 0.74-0.87, reliability with 0.62-0.89 and validity in comparison to the Child Behavioural Checklist (CBCL) with 0.75-0.87[19]. Nevertheless, the SDQ shows a shortcoming in detecting PTBS and therefore has to be applied with caution among ASRM. Considering existing limitations, validated, easy-to-use tools are still missing[16,17].

Even when mental health needs are recognized, adequate care is rarely ensured. In Germany, only 12% of adult ASR received minimally adequate therapy[20]. Carbonell et al. established categories for barriers to mental health care for children and adolescents in general: structural and financial barriers, attitudes to treatment, professional interventions and shortcoming of the biomedical model[14]. Dumke et al. identified seven categories for barriers to mental health care for ASR adults in high-income countries: patients understanding of mental illness, fear of stigma, lack of awareness of services, attitudes to treatment, language barriers, practical and structural issues and providers´ attitudes and competences[13]. Such barriers intersectionally affect the group of ASRM. Additionally, fragmented care structures hamper access[14,21], while interprofessional teamwork and local networks could promote more integrated and patient-centered care[21,22]. How inter-organisational collaborations across different sectors involved in child mental health care are perceived and transferred to practice is complex and depends on individual, group and organizational aspects[23]. Language barriers further complicate access, particularly in mental health[6,24]. Yet, recent data on how pediatric professionals address communication issues in Germany is lacking.

This study explores multiprofessional experiences of HSCPs in caring for ASRM regarding mental health problems and communication challenges, focusing on: (1) perceived burden of mental health problems, (2) screening and follow-up practices, (3) impact of communication problems. We hypothesized that ASRM show relevant mental health needs, but screening and follow-up are rarely implemented, and communication problems remain highly prevalent despite diverse but insufficient strategies. In our SAVE-KID study, 21.2% of ASRM are reported to have mental health problems, consistent with previous findings. Regular mental health screening is largely not implemented, and only 37.1% of affected cases receive follow-up care. Most HSCPs report frequent communication barriers, which impair care in over 80% of cases.

## Methods

The cross-sectional survey was conducted between September 2023 and February 2024. Data were collected and managed using REDCap electronic data capture tools hosted at the University Hospital Freiburg, Germany[25,26]. The target group were professionals and volunteers working with ASRM in a health-related context in Germany such as medical doctors, social workers, psychologists, interpreters, and state employees at the social welfare office, regional councils or reception centers. ASRM were defined as displaced children and adolescents under 18 years who arrived in Germany over the last two years in search of protection, accompanied or unaccompanied by family members. Participants were recruited by email via professional societies (Infectious Diseases Societies (German Society for Pediatric Infectious Diseases (DGPI), German Society for Infectious Diseases (DGI)), Tropical Medicine Societies (Society for Tropical Paediatrics and International Child Health (GTP) e.V., German Society for Tropical Medicine, Travel Medicine and Global Health (DTG) e.V., Pediatric Societies (German Society of Pediatrics and Adolescent Medicine (DGKJ) e.V., Berufsverband der Kinder- und Jugendärzt*innen (BVKJ) e.V.) and via multipliers (e.g., Caritas, German Red Cross (DRK) e.V., PriCareNet research network). After accessing the study link, participants were presented with the informed-consent document and asked to indicate consent by selecting a checkbox and clicking the "Proceed" button.

### SAVE-KID survey

The survey included three thematic areas: experiences with mental health problems in ASRM, communication and infectious disease screening. To access the mental health status and care we asked for applied screening, observed symptoms, and referral to follow-up or therapy. We examined how the impact of language and communication issues is perceived in a language-discordant setting, and which support strategies are applied in practice. Regarding infectious disease screening, we asked for the frequency of conducted structured screening, which examinations were applied as well as for offered vaccinations comparing to the national guidelines. The survey was conducted in German, an English translation has been annexed to the supplements (S1).

The survey contained general and profession/working place-specific questions in a mostly closed, Likert-scaled format. A Likert scale with six answer options ("always", "very frequent", "frequent", "occasionally", "rarely", "never") was applied with an opt-out answer ("unknown", "does not apply"). Most sections were complemented by optional text fields. To access perception of psychological and psychosomatic symptoms, we asked for the frequency of observed or reported symptoms and behaviors in the cohort of seen ASRM. Based on the different versions of Strength and Difficulties Questionnaire (SDQ[9]) for age-groups, self-, parent- and teacher-reports, we chose 16 common key words. Positive attributes from the prosocial behavior scale were not included. For analysis, the items were further clustered into internalizing symptoms (aggressiveness towards oneself, apathy, lack of concentration, lonerism, nightmares, sadness, sleep problems, tiredness), externalizing symptoms (aggressiveness towards others, hyperactivity, impulsiveness, irritability, restlessness) and somatic complaints (recurrent headaches or abdominal pain)[27]. Additionally, drug and alcohol use were queried. The survey was developed within a multi-professional team of medical doctors experienced in in- and out-patient care, a psychologist, and medical students. Three external medical doctors with extensive experience in ASRM treatment reviewed the questionnaire for comprehensibility, length, and relevance to their daily practice and their feedback was implemented.

The contexts of the contacts to ASRM and therefore for the behavioral observations are very heterogeneous and inherit potential biases. For example, minors who are presented to medical staff due to an acute infectious disease will prompt a different focus of the staff than a presentation for a planned developmental screening. As the totality of our cohort is unknown, prevalences are difficult to estimate. Also, individual ASRM could be reported multiple times by several professional groups (e.g., from the same shelter). Yet, this heterogeneity seemed the closest to the reality that is encountered by health and care professionals in practice.

**Table 1 | Sample characterization**

| Category | Variable | All participants (n = 201; 100%) | Medical doctors (n = 89; 44.3%) | Social workers (n = 76; 37.8%) | Others (n = 36; 17.9%) |
|---|---|---|---|---|---|
| Gender | Female | 147 (73.1%) | 52 (65%) | 59 (78%) | 30 (83%) |
| | Male | 54 (26.9%) | 28 (35%) | 17 (22%) | 6 (17%) |
| | Diverse | 0 | 0 | 0 | 0 |
| Age | Mean | 44.4 y | 50.1 y | 37.7 y | 44.6 y |
| | Min | 18 y | 29 y | 24 y | 18 y |
| | Max | 73 y | 72 y | 58 y | 73 y |
| Working place | Municipal shared accommodation | 39 (19.4%) | 5 (6.25%) | 27 (35.5%) | 7 (19.4%) |
| | Medical practice | 39 (19.4%) | 30 (37.5%) | - | 4 (11.1%) |
| | State reception center | 38 (18.9%) | 7 (8.8%) | 27 (35.5%) | 4 (11.1%) |
| | Administration | 27 (13.4%) | 13 (16.3%) | 6 (7.9%) | 8 (22.2%) |
| | Hospital | 17 (8.5%) | 15 (18.8%) | - | |
| | Other | 41 (20.4%) | 10 (12.5%) | 16 (21.1%) | 13 (36.2%) |
| Working experience (at current working place) | Mean | 8.6 y | 12.6 y | 4.6 y | 7.4 y |
| | Min | 0.5 y | 0.5 y | 0.5 y | 1 y |
| | Max | 33 y | 33 y | 14 y | 30 y |

Participants self-identified their gender as female, male or diverse. *n* number, *y* years.

**Table 2 | Numbers of ASRM seen in consultation, with observed mental health problems and receiving follow-up (n$_{(respondents)}$ = 167)**

| Statistical parameters (referring to last three months) | ASRM in consultation | ASRM with mental health problems | ASRM with mental health problems and referral/follow-up |
|---|---|---|---|
| Total | 13,096 | 2757 | 905 |
| Mean | 78.4 | 16.5 | 6.1 |
| Median | 30 | 10 | 3 |
| Min–Max | 0−1026 | 0−250 | 0−190 |
| Q1–Q3 | 10−95 | 5−20 | 1−5 |

*Q1* first quartile, *Q3* third quartile.

## Statistical analysis

As the survey consisted of three distinct areas, we were able to analyse the survey parts separately. However, a participant's answer was only included if the respective survey part (infectious disease screening, communication, mental health) was completed.

Statistical analysis was performed using SPSS (IBM Statistical Package for Social Science Statistics 29) and Excel (Microsoft Excel, Version 2411). Frequencies, mean, median, minimum and maximum values and interquartile ranges were calculated for the totality and for subgroups. As for the small total numbers and heterogeneous data a comparison of subgroups was limited to the largest subgroups. Differences between subgroups were reported as relevant tendency when reaching ≥ 10%. Whenever indicated we conducted tests of significance (t-test, $\chi^2$ test). Free texts were summarized under keywords according to content agreement and cross-checked by three of the authors. Graphics were generated using the R (version 4.3.1) and the package ggplot2 (version 3.4.2).

## Results

### Participants

201 participants took part in the survey (73.1% female, 26.9% male). 44.3% were medical doctors, 37.8% social workers, and 17.9% other professionals. 76.4% of the physicians worked in pediatrics. Participants worked in communal shared accommodations (19.4%), state reception centres

(18.9%), medical practices (19.4%), administration (13.4%), hospitals (8.5%), and other organisational structures such as counselling centres, non-governmental and state organisations. In total, participants from 13 of the 16 German federal states were included, 59.7% of responses were from Baden-Wuerttemberg, 16.9% from Bavaria. The mean age of participants was 44 years (min 18 y.; max 73 y.), with a mean professional experience of 8 years (min 0.5 y., max. 33 y.).

181 participants answered the questions on communication and language barriers. 167 participants completed the section on mental health assessment. If not indicated differently (due to filters related to previous questions), percentages refer to n = 181/167 as 100.0% respectively. Characteristics in both subsamples did not differ relevantly compared to the general cohort (Δ < 5%). Wherever indicated, we present results separately for the largest professional groups (medical doctors and social workers, see Table 1).

**Mental health (n$_{(all)}$ = 167; n$_{(medical doctors)}$ = 76, n$_{(social workers)}$ = 62, n$_{(others)}$ = 29, n$_{(missing)}$ = 34)**

Participants were asked to specify the number of ASRM seen in their institution/practice over the last three months. Next, we specifically focused on mental health problems, and how many of those ASRM noted with mental health problems received referrals or follow-up care (see Table 2). In total, 13,096 contacts with ASRM over the last three months were estimated, which corresponds to an average of 78.4 contacts per respondent. 21.2% (total 2757, mean 16.5 per respondent) presented with one or more listed mental health problems. Of those 37.1% (total 905, mean 6.1 per respondent) received follow-up care. If there was any mental health screening used within the institution, regardless the frequency, more ASRM were reported with mental health problems (26.0% vs. 17.8%). Also, a higher percentage received follow-up care (40.2% vs. 25.2%). There were no relevant changes between respondents, who had undergone further training in mental health care or not. There were no significant differences between medical doctors and social workers in their estimated number of ASRM with mental health problems seen ($t(136) = -0.571$, $p = 0.569$) or referred ($t(122) = -0.830$, $p = 0.408$).

**Screening on mental health.** To address mental health assessment, we investigated the frequency of conducted screening on mental health problems among ASRM. 46.1% of respondents reported that their

institution had no screening procedure at all. 16.7% reported that they conducted screening rarely and 12.0% at best occasionally, 13.8% did not have information on procedures in their institution. 11.4% reported (very) frequent screening. Furthermore, 53.3% indicated that findings on mental health problems were at best occasionally documented. Slight differences between professional groups showed, that more medical doctors reported implemented screening and documentation within their institution than social workers (11,8% vs. 6,5% and 31,6% vs 21,0%, respectively), likely due to distribution of tasks. These differences are not significant ($\chi^2(1) = 1.16$, $p = 0.281$ for screening; $\chi^2(1) = 1.96$, $p = 0.162$ for documentation.

Answers regarding applied screening tools for mental health problems included informal questions during consultation, interviews by psychologists or other trained staff as well as the use of following questionnaires in original or adapted versions: Strength and Difficulties Questionnaire (SDQ[9]); Refugee Health Screener 15 (RHS-15[18]); Mannheimer Eltern-Fragebogen (MEF[28]); Diagnostik-System für Psychische Störungen im Kindes- und Jugendalter (DISYPS-II[29]), Clinician-Administered PTSD Scale (CAPS[30]).

**Observed mental health status of ASRM.** Based on the SDQ items, we collected information on mental health problems and respective frequencies observed among ASRM. Frequent issues were lack of concentration (reported as "always", "very frequently" or "frequently" in 55.1%), restlessness (53.3%), tiredness (43.1%), and sleep disorders (43.1%). In contrast, the least observed issues were aggressiveness towards others (14.4%) or oneself (10.6%) or drug and alcohol abuse (6.6%). Some symptoms were more difficult to estimate or were less often asked about. Herein items most often reported as "not assessable" were nightmares (36.5%), recurrent headaches or abdominal pain (31.7%) or drug and alcohol abuse (37.7%). Additional issues and frequencies are shown in Fig. 1. Yet, there were differences if screening was implemented as well as between professional groups. We clustered internalizing and externalizing symptoms as well as somatic complaints that may have a psychosomatic etiology. Both, internalizing and externalizing clusters, were equally reported with 35.9% and 34.3%, respectively. (Psycho-) Somatic complaints were reported a little less frequent with 27.2%. Participants working in institutions, where some type of mental health screening was implemented, regardless the frequency, reported more internalizing and psychosomatic complaints as participants from institutions where no screening at all was implemented or no information on screening implementation was available (internalizing symptoms: 46.1% vs. 31.6%; psychosomatic complaints: 36.6% vs. 21.5%). Meanwhile, externalizing symptoms were described almost equally (31.1% vs. 36.4%). To a great extent, differences were traced back to a less frequent response "unknown" within the screening group. Medical doctors noted higher frequencies of recurrent headaches or abdominal pain and sadness. Meanwhile, social workers observed more restlessness, impulsiveness, and hyperactivity as well as irritability, aggressiveness towards others and lack of concentration.

**Resources and barriers in mental health care.** 72.5% claim that on a regular basis they lack sufficient resources to screen, initiate further diagnostics, and follow-up on children and adolescents with mental health problems. Medical doctors perceived this lack of resources even more drastically than social workers (81,6% vs. 69,4%). However, this difference is not significant ($\chi^2(1) = 3.57$, $p = 0.06$). Most reported barriers for follow-up were lack of availability of appointments (86.1%), lack of availability of interpreters (79.1%), and lacking cost-coverage (62.0%). While the first mentioned rather refers to mental health care structures in general, the latter two barriers are specific to our cohort. Text field answers showed that complex infrastructures with relocation processes and unclear task distribution in inter-institutional cooperation seem to impair adequate follow-up. Further comments mentioned shortage of professional staff, lack of expertise/experience, and rejection of refugees

by institutions or psychotherapists due to language barriers or other reasons (e.g., general decision not to work with interpreters). Participants noted that the challenge lies not in the refusal of cost coverage by the responsible institutions but in the complicated and time-consuming application process, which hinders the use of existing resources. 60.5% indicated that the access to resources for additional mental health diagnostics, therapy, and follow-up was considerably more difficult for ASRM compared to German residents.

**Dealing with challenges in mental health care.** Participants were asked to subjectively evaluate whether they could meet the perceived needs of the ASRM within their professional role to ensure successful (mental health) care. 50.0% ($n = 148$) of participants reported that they felt they could rarely or never meet the needs of ASRM; 33.1% stated to feel able to meet their needs at best occasionally. This did not relevantly differ between professions, training status or working experience ($\leq 5$ y vs. $>$ 5 y). 23.7% indicated that they frequently to always felt overwhelmed when dealing with ASRM with mental health problems; 52.7% felt occasionally overwhelmed. This item was influenced by having received some sort of professional training on working with minors with mental health problems (14.0% with training vs. 28.6% without training felt frequently to always overwhelmed). 66% of participants lack a respective training. Social workers significantly more often underwent a professional training on working with minors with mental health problems compared to medical doctors (46.8% vs. 26.3%, $\chi^2(1) = 6.24$, $p = 0.012$).

**Communication ($n = 181$, $n_{(medical\ doctors)} = 82$, $n_{(social\ workers)} = 68$, $n_{(others)} = 31$, $n_{(missing)} = 20$)**

84.0% reported encountering communication problems at least frequently when consulting ASRM. Of all participants who had experienced communication problems ($n = 167$), 81.4% indicated that those issues at least frequently impaired sufficient health care measures (75.1% of total respondents). Medical doctors stated significantly more communication problems (90,2% vs. 63,4%; $\chi^2(1) = 4.59$, $p = 0.032$), meanwhile impairment was equally reported. 45.9% reported to (very) frequently lack communication aids. Wherever communications aids were available, the most used aids were online tools (e.g., apps like Google translate, SayHi), relatives, and interpreters in presence, with 71.3%, 55.8%, and 47.0%, respectively. In addition, interpreters via telephone and staff members were often involved (Fig. 2). Meanwhile, experience and use of video-interpreters were very limited (6.1%). Further comments in text fields included pictograms, pre-translated information sheets, translation devices (e.g., Vasco translator), mimics and gestures as well as other inhabitants of the shared accommodations or reception centers.

Also, answers on barriers to mental health care reflected the importance of assured communication for access to health care. 79.1% agreed that the lack of interpreters impairs follow-up on affected ASRM. In free text fields, participants stated repetitively that health-care institutions may reject ASRM because of lacking language skills. According to the respondents, the time-consuming application and unsure response on the cost-coverage for interpreters further limits access to mental health care in practice.

## Discussion

SAVE-KID outlines the mental health situation of ASRM in Germany. On average, 21.2% were reported with mental health problems. Yet, < 24% of HSCP implemented regular screening, and only 37.1% of cases received follow-up. 84% of HSCP reported frequent communication problems, in > 80% impairing care. Most common aids were online tools and relatives.

ASRM show high vulnerability to mental health issues, shaped by social networks, flight history and legal status. We found 21.2% prevalence and higher where screening was used. These results align with Janda et al. (24.6% of unaccompanied refugee minors (URM) presented with mental health problems)[31]. Systematic reviews and SDQ studies indicate that ASRM relocated to Western countries often experience even higher prevalences of mental health problems, particularly PTSD (25.6–62.7%)[7,32]. Two studies

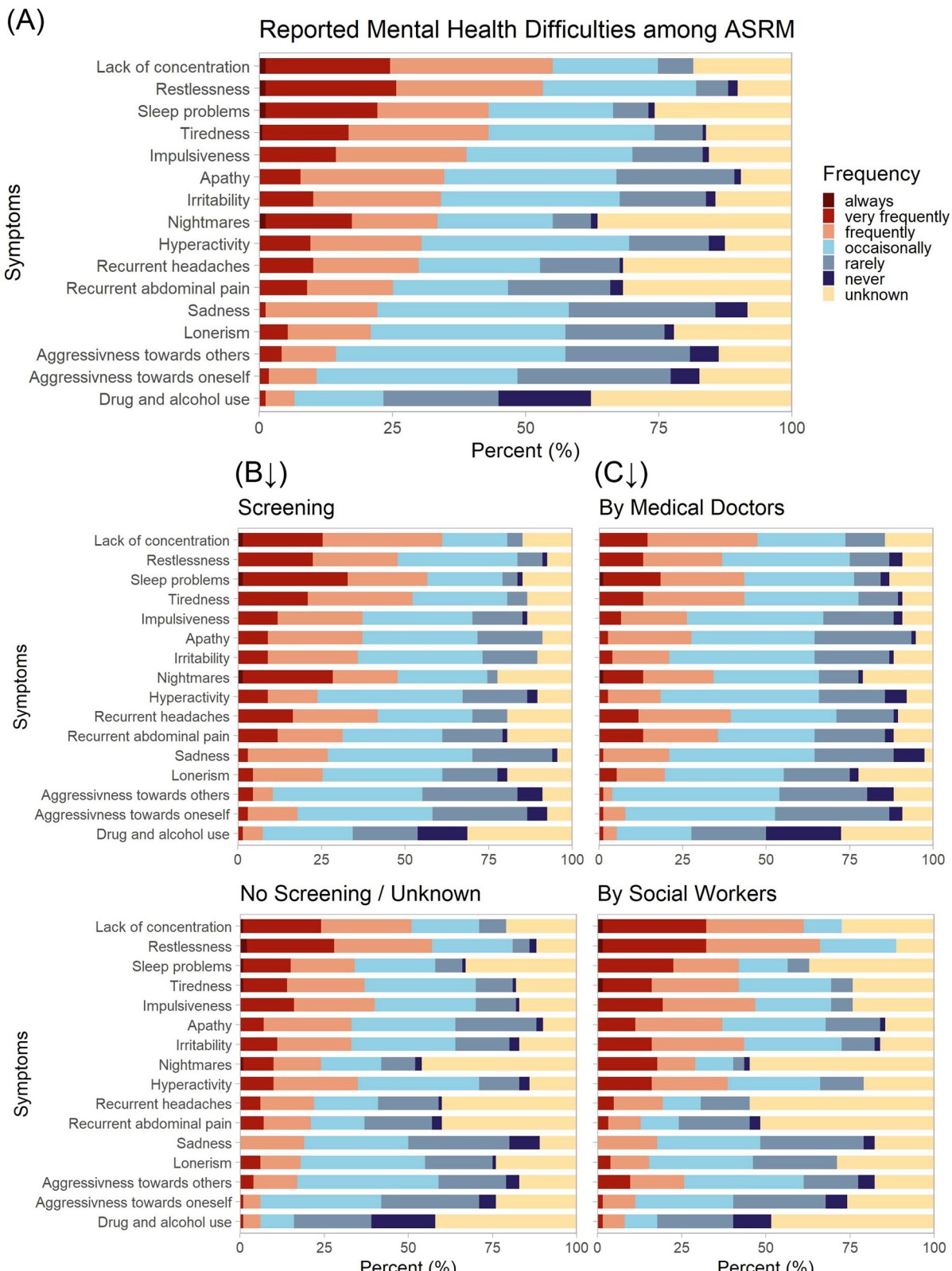

**Fig. 1 | Reported mental health problems among ASRM. A** Overall reported symptoms sorted by frequencies. Answer categories are depicted in colours and shown in percent ($n_{(all)}$ = 167). **B** Symptoms separately reported for participants from institutions with (infrequent) screening ($n_{(screening)}$ = 67) vs. no or unknown screening implementation ($n_{(no/unknown\ screening)}$ = 100). **C** Symptoms separately reported by medical doctors ($n_{(medical\ doctors)}$ = 76) and social workers ($n_{(social\ workers)}$ = 62).

**Fig. 2 | Reported frequency of applying different communication aids in health care for ASRM.** Answer categories are depicted in colours and shown in percent (n(respondents) = 181).

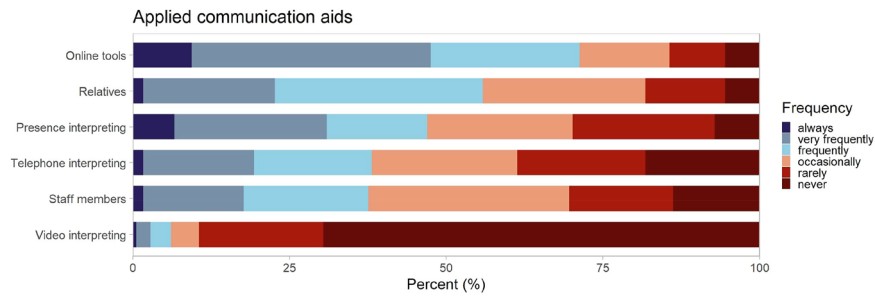

applying the SDQ in Sweden and Denmark report 25% and 31% of ASRM scoring above the cut-off for relevant mental health problems[33,34]. Especially vulnerable groups such as URM are regularly reported as more frequently affected[35]. Meanwhile, rates among their German-speaking peers range between 14.5% and 21%[10–12]. The COPSY study showed an increase up to ~31% during COVID-19[36], with a slight decrease since 2021, yet around one in four minors still affected[37]. Comparability of the data is limited due to methodological heterogeneity. Our method - based on HSCP reports, in a context of no regularly implemented mental health screening, and with the SDQ-based questions of our questionnaire potentially underestimating PTSD - likely underestimates prevalence. Meanwhile, lack of diagnostic thresholds, as no specific clinically relevant cut-off or no confirmed diagnosis were requested, may lead to overestimation. Still, results indicate ASRM face comparable or higher mental health burden than peers in Germany.

Some profession-related differences emerged. Social workers reported more externalizing, physicians more (psycho-)somatic symptoms such as recurrent pain. This might reflect both setting and training foci. Interprofessional assessment and self-report instruments are crucial, especially for internalizing problems that are often unnoticed by caregivers[38,39].

Screening remains inconsistent: almost half reported no standardized procedure, more than half no regular documentation. Nevertheless, several screening tools (e.g., SDQ, RHS-15, MEF, DISYPS-II, CAPS) have already been established in practice, albeit only sporadically. Teams and institutions frequently working with ASRM should evaluate their use within the specific context and establish standard procedures for screening, documentation and follow-up. Regular documentation is crucial for data quality assurance and care continuity within the context of relocation processes and fragmented systems. There is no national recommendation about when best to screen ASRM for mental health problems. We advocate for early screening post-arrival, before relocation complicates access. Research on effective, validated tools for ASRM is needed and institutions should define standard procedures- whenever possible along with implementation monitoring. Multicentric studies are needed to generate representative data. Despite the lack of scientific evidence, we recognize a need for a medical guideline in practice. Recommendations on mental health assessment, communication, and social pediatric measures for ASRM in Germany would offer orientation for professionals and institutions, standardize procedures, and underline political demands.

Furthermore, training opportunities for HSCP should expand, while existing programs, enablers, and barriers need to be further evaluated. Training programs for social workers may open up to medical staff and herein offer opportunities for interdisciplinary learning and networking. Subsequently, self-efficacy of professionals may improve while stress factors may decrease. Where no programs on working with minors with mental health problems or in language-divergent and intercultural contexts are available, professional associations are called into action.

Of our cohort, only 37.1% of affected ASRM received follow-up - similar to German-speaking cohorts (21–27%), underlining the lack of information on objective and perceived treatment needs as well as on appropriateness of treatment[10,11]. Despite a relatively well-funded (mental) health care system and high inpatient capacities, Germany exhibits low

numbers of child and adolescent mental health specialists as well as outpatient capacities leading to long waiting times and undertreatment[37]. This suggests systemic underutilization of care resources, not only ASRM-specific gaps. Still, structural, financial, and provider-related barriers weigh heavily and are potentially aggravated for ASRM. In comparison to Dumke et al.[13] and Carbonell et al.[14], HSCPs in our study highlighted various barriers: shortage of appointments (structural barrier) and interpreters (language barrier), rejection by institutions (providers´ attitudes) due to limited language skills and/or time-restraints as well as complicated or denied cost coverage (financial barrier). Also in the German context, trained staff is scarce (providers' competences). Social workers emphasized financial and organizational issues; whereas physicians emphasized access to interpreters. In practice, structural knowledge on available support options is crucial, yet difficult to assess for HSCPs who interact only intermittently with ASRM. Not only is there a lack of awareness of services on the demand-side, but also on the providers´ side in the context of a highly fragmented German health care system[40]. Providers' attitudes and transcultural competences are addressed in training initiatives (e.g., medical faculties, nursing associations), though coverage and sustainability vary. Strengthening local and interprofessional networks can help share knowledge and improve access. A possible follow-up project could involve measuring the perception and quality of interprofessional collaboration using the PINCOM framework and questionnaire[23]. Selected model projects, such as the Konstanz program for coordinated psychotherapeutic care of refugees (KOBEG) or the Munich initiative Intercultural Child Development (INCLUDE), illustrate how structured and interprofessional approaches can improve support. KOBEG implemented a central coordination office and trained "health peers" from the community[41]. INCLUDE offers culture- and trauma-sensitive care for children from refugee families via interprofessional consultation, parents' college, and employee trainings[42]. Another way of expanding social support structures would be to draw on the existing support structures of Early Childhood Intervention[43], including the interpreting services used, and to open these up to ASRM beyond the age of three. Scaling up and evaluating such approaches could inform future care structures.

Our research addresses the early stage of ASRM interaction with the German health care system shortly after arrival and usually before acquiring sufficient language skills. Therefore, language barriers are omnipresent and frequently perceived as a relevant impairment to adequate care. Online tools are the most used aids, free of charge and available anywhere and anytime, yet enabling only very limited communication and no control over the translated content's accuracy. Relatives, including children, frequently translate. ~50% of HSCPs are experienced in working with in-person interpreters, but video-translation remained rare and only available in specific settings (e.g., university hospitals). Studies report that working with both relatives and professional interpreters has distinct advantages and drawbacks[24,44]. While professional and cultural-sensitive interpreting may enhance trust and accuracy, challenges around confidentiality remain. Further professional trainings need to target adequate application of available communication aids. Future studies should explore video-interpretation acceptability of all stakeholders (families and HSCP). Potential change of these services' cost-coverage, application processes and availability is a urgent political question.

Limitations of our survey include lack of information on the overall HSCP population as well as the ASRM who are (or are not) engaged with care structures. Due to this, and to methodological heterogeneity, no true prevalence rates could be calculated and comparability with other studies is limited. Evaluation of subgroups was restricted by small sample sizes. National representativeness is limited, especially considering the variability of care structures within the German federal system[40]. Most participants (76.6%) were from southern Germany (Baden-Wuerttemberg, Bavaria), where most ASR cross the German boarder and still around 30% are accommodated based on federal law (AsylG §45). Recruitment primarily involved professionals affiliated with infectious diseases associations, likely influencing both expertise and perspective within the sample. Besides, we did not collect data on relevant characteristics of ASRM such as age, country of origin, flight history, legal status, or whether they came (un-)accompanied. Some estimates provided by participants were difficult to obtain or interpret - heterogeneous responses may partly reflect the diversity of care contexts but also challenges in reporting.

Despite these limitations, retrospective assessment by a broad group of HSCP enabled easy access to a diverse overview of ASRM's mental health status as perceived in routine care. This multiprofessional perspective, spanning various care settings, is a key strength of our study, adding valuable insight into a complex reality. With nearly 200 participants and around 13,000 ASRM contacts over the last three months, this work contributes important data on current mental health burden and care for ASRM in Germany and highlights directions for needed service. Future research should include the perspectives of ASRM and their families from the earliest stages - both in developing research questions and during data collection - to better understand how culturally sensitive screening and follow-up can be effectively implemented. In this regard, a longitudinal study could provide further insights into the development and course of mental health problems among ASRM. Such work would require a different methodological approach, including direct assessments of ASRM rather than proxy reports.

## Conclusion

Annually, around 300,000 minors from other countries arrive in Germany[3] seeking and entitled to the opportunity for optimal development and well-being (SDG 3, CRC 24). SAVE-KID reveals that a substantial proportion of ASRM is affected by mental health problems and requires specialized health care services. Despite the lack of structured screening, HSCP report mental health problems in over 20% of cases. Ongoing challenges include communication barriers, insufficient specialized staff, and the lack of practical, validated screening tools - factors that together hinder comprehensive identification of mental health needs. Furthermore, appropriate follow-up care is often lacking, with complicated cost-coverage procedures compounding access difficulties. From a professional perspective, adequate psychosocial support for ASRM remains frequently unavailable. Broad implementation of systematic screening is urgently needed to ensure timely access to medical and therapeutical care. Trained staff and expanded, accessible follow-up care structures must be ensured. Local, scientifically supervised projects may serve as role models, but need to become established and permanent to ensure sustainable care.

## Ethics statement

Ethical approval was obtained by the Ethics Committee of Albert-Ludwigs-University Freiburg on 08/23/2023.

## Data availability

The de-identified dataset generated and analyzed during the current study is made available as dataset in the supplements.

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

## Acknowledgements

The authors want to acknowledge the contribution of Dr. C. Adam, Dr. R. Fressle, Dr. A. Krieg, Dr. V. Boda and Dr. K. Kneser for reviewing the questionnaire and to thank Ayfer Sen from the Ombudsstelle für Flüchtlingserstaufnahme Baden-Württemberg for distributing the questionnaire and supporting the study team. We thank the Pediatric Healthcare Research Group as platform for discussion and for their valuable feedback at different stages of the study. This research was funded by the Theodor-Escherich-Prize 2023 of the German Society for Pediatric Infectious Diseases (DGPI) granted to B.S.

## Author contributions

A.E., B.S., and J.W. conceived and designed the study. B.S. and J.W. conducted the recruitment and data collection. A.E. and J.W. analysed the data. A.E. and B.S. interpreted the results. A.E. wrote the initial manuscript draft. M.K., T.L., and M.H. contributed across all stages of the study by discussion and content-related input. All authors commented on, reviewed, and approved the final manuscript.

## Funding

## Competing interests

The authors declare no competing interests.
