## [Transparent Peer Review file · Communications Medicine]

Cross-sectional survey among professionals on communication and mental health care for asylum seeking and refugee minors in Germany

Corresponding Author: Dr Benedikt Spielberger

Version 0:

Reviewer comments:

Reviewer #1

(Remarks to the Author)

The manuscript "SAVE-KID: Cross-sectional survey among professionals on communication and mental health care for asylum-seeking and refugee minors in Germany" presents a cross-sectional quantitative study that describes the experiences of health and care professionals working with asylum-seeking and refugee minors in various contexts in Germany. This is a relatively under-researched group that would benefit from increased attention and evidence-based insights.

However, the manuscript could be improved before being considered for publication. The abstract lacks key details, such as the total number of participants (sample size) and their sociodemographic characteristics. While the introduction is brief—understandable given the limited existing evidence on this topic—it could be expanded using key findings from the study. Concepts such as "fragmentation of care," "person-centered care," "multidisciplinary teamwork," and "integrated care" emerge as important themes in the results and could serve as a foundation for a more developed theoretical background.

The discussion section would also benefit from a deeper reflection on these concepts, as well as the inclusion of practical recommendations for professionals working with asylum-seeking and refugee minors. Exploring new methodologies, such as the implementation of case managers or multidisciplinary team discussions for complex cases, could provide solutions for improving care coordination, reducing gaps in follow-up, and addressing transdisciplinary biases.

The methods usually are after the introduction and they are more clear when using sub-titles. The level of detail is adequate.

Reviewer #2

(Remarks to the Author)

Version 1:

Reviewer comments:

Reviewer #1

(Remarks to the Author)

Thank you very much for integrate everything.

Reviewer #2

(Remarks to the Author)

I have carefully reviewed the revised version of your manuscript. First of all, I would like to sincerely congratulate the

research team for the enormous effort made in this revision. It is not common to find such a careful, clear, and well-structured response to the reviewers' report. The work shows meticulous attention not only in incorporating the changes but also in justifying each decision made. This level of dedication deserves recognition and reflects a serious commitment to improving the manuscript and to the scientific review process.

The manuscript, in its current version, represents a valuable and relevant contribution in a field where empirical data remain scarce and underinterpreted. Nevertheless, I believe there are still some aspects that could be improved before its final acceptance. I will be brief...

First, the discussion could more explicitly reinforce the applicability of the findings in terms of professional practice and public policy development. While the authors' methodological caution in not deriving direct recommendations from a cross-sectional study is understandable, it would still be possible to provide a synthesis of general lessons transferable to other contexts, drawing on the local projects mentioned. In addition, the authors could state more clearly the practical, policy, and research recommendations that emerge from the study, thereby guiding future lines of work. Researchers need studies not only to provide data but also to point out what needs to be addressed in the future. A greater degree of interpretation is therefore requested.

Second, the comparative analyses between physicians and social workers have been incorporated, as previously suggested, although they are presented in a rather brief manner. It would be advisable to specify them in greater detail and add a brief—very brief—interpretation of their possible implications for both professional practice and training.

Finally, although the references have been updated and expanded, it would be advisable to incorporate, if available, publications from 2025 on mental health policies in Germany that would strengthen the contextualization of the study.

In summary, I would like to once again thank the authors for the work carried out and the improvements introduced. The manuscript has been substantially enriched and constitutes a solid and relevant contribution in the field of mental health. My observations are limited to minor adjustments, which I am confident the authors will be able to address successfully.

Version 2:

Reviewer comments:

Reviewer #2

(Remarks to the Author)

I would like to thank the authors for the effort made in addressing and discussing each of my comments. I congratulate them on the excellent work done, and it is with great pleasure that I accept the manuscript.

Dr. Benedikt D. Spielberger
Division of Pediatric Infectious Diseases and Rheumatology
& Institute for Infection Prevention and Control
University Medical Center Freiburg
Email: benedikt.spielberger@uniklinik-freiburg.de

Editorial Office
Communications Medicine

Freiburg, Germany
November 9, 2025

Dear Ladies and Gentlemen,

Please find attached the revised version of our manuscript, "*Cross-sectional survey among professionals on communication and mental health care for asylum seeking and refugee minors in Germany*", submitted for publication as an Original Article in *Communications Medicine*.

We have addressed the reviewer comments and believe that these changes have strengthened the clarity and rigor of the manuscript.

Our concise, point-by-point response is enclosed.

Reviewers' comments:

Reviewer #1 (Remarks to the Author):

Thank you very much for integrate everything.

Reply: We thank Reviewer #1 for their support and are delighted to have met your expectations.

Reviewer #2 (Remarks to the Author):

I have carefully reviewed the revised version of your manuscript. First of all, I would like to sincerely congratulate the research team for the enormous effort made in this revision. It is not common to find such a careful, clear, and well-structured to the reviewers' report. The work shows meticulous attention not only in incorporating the changes but also in justifying each decision made. This level of dedication deserves recognition and reflects a serious commitment to improving the manuscript and to the scientific review process.

The manuscript, in its current version, represents a valuable and relevant contribution in a field where empirical data remain scarce and underinterpreted. Nevertheless, I believe there are still some aspects that could be improved before its final acceptance. I will be brief..

First, the discussion could more explicitly reinforce the applicability of the findings in terms of professional practice and public policy development. While the authors' methodological caution in not deriving direct recommendations from a cross-sectional study is understandable, it would still be possible to provide a synthesis of general lessons transferable to other contexts, drawing on the local projects mentioned. In addition, the authors could state more clearly the practical, policy, and research recommendations that emerge from the study, thereby guiding future lines of work. Researchers need studies not

only to provide data but also to point out what needs to be addressed in the future. A greater degree of interpretation is therefore requested.

Reply: We hope to meet this demand for practical applicability of the results by making further additions. At the individual and institutional level, we have highlighted the possibility of applying the screening instruments used by several of our respondents and develop standards of documentation to facilitate continuity. In addition to the aforementioned goal of building up local interprofessional networks we added the proposal of drawing back on the Frühe Hilfen (Early Childhood Intervention), where interprofessional networks already support challenged families and an interpreter pool is established.

Reviewer #2

Second, the comparative analyses between physicians and social workers have been incorporated, as previously suggested, although they are presented in a rather brief manner. It would be advisable to specify them in greater detail and add a brief—very brief—interpretation of their possible implications for both professional practice and training.

Reply: Comparative analyses between the professional groups were supplemented by further details and tests of significance (p. 6, p. 8). We amplified the recommendations on professional training towards using existing structures and knowledge to scale up training programs for interprofessional learning.

Reviewer #2

Finally, although the references have been updated and expanded, it would be advisable to incorporate, if available, publications from 2025 on mental health policies in Germany that would strengthen the contextualization of the study.

Reply: In our search for the latest information on relevant measures in the field of mental health in Germany, we included the review on outpatient psychotherapeutic care for children and adolescents in Germany (Rodney-Wolf and Schmitz, 2025).

Reviewer #2

In summary, I would like to once again thank the authors for the work carried out and the improvements introduced. The manuscript has been substantially enriched and constitutes a solid and relevant contribution in the field of mental health. My observations are limited to minor adjustments, which I am confident the authors will be able to address successfully.

Reply: We would like to thank reviewer#2 for his valuable suggestions and hope that our changes meet the reviewers' expectations.

We trust these revisions resolve the remaining concerns and have improved our manuscript. We thank both reviewers for their critical and helpful contributions. We look forward to your feedback.

Yours sincerely,

Anna J. Esser, Jana Willems and Dr. Benedikt D. Spielberger
